## Replications: stage 2

cognition/psychology

social evaluation, infant cognition, manual choice paradigm, moral judgement, partner choice

**Author for correspondence:**
Laura Schlingloff
e-mail: schlingloff_laura@phd.ceu.edu

# Do 15-month-old infants prefer helpers? A replication of Hamlin *et al.* (2007)

Laura Schlingloff[1], Gergely Csibra[1,2] and Denis Tatone[1]

[1]Cognitive Development Center, Department of Cognitive Science, Central European University, Budapest, Hungary
[2]Department of Psychological Sciences, Birkbeck, University of London, London, UK

LS, 0000-0001-7512-2617; GC, 0000-0002-7044-3056; DT, 0000-0001-6694-2656

Hamlin *et al.* found in 2007 that preverbal infants displayed a preference for helpers over hinderers. The robustness of this finding and the conditions under which infant sociomoral evaluation can be elicited has since been debated. Here, we conducted a replication of the original study, in which we tested 14- to 16-month-olds using a familiarization procedure with three-dimensional animated video stimuli. Unlike previous replication attempts, ours uniquely benefited from detailed procedural advice by Hamlin. In contrast with the original results, only 16 out of 32 infants (50%) in our study reached for the helper; thus, we were not able to replicate the findings. A possible reason for this failure is that infants' preference for prosocial agents may not be reliably elicited with the procedure and stimuli adopted. Alternatively, the effect size of infants' preference may be smaller than originally estimated. The study addresses ongoing methodological debates on the replicability of influential findings in infant cognition.

## 1. Introduction

A growing literature suggests that, from a very young age, infants spontaneously engage in third-party social evaluation, drawing inferences about the sociomoral dispositions of unrelated agents on the basis of their interactions with others. This proliferating research project was launched by the seminal 2007 study of Hamlin *et al.* [1], which showed that 6- and 10-month-olds presented with two characters interacting in a helpful or harmful manner towards a common patient subsequently preferred the former when prompted to choose among the two.

This study, and others that followed in its wake, bolstered the empirical case for an 'innate moral core' [2]: an early-developing set of abilities that allows infants to infer sociomorally relevant dispositions from the behaviour of third parties, which in turn enables infants to recognize and selectively interact with

potentially cooperative partners. Support for this account has been found in a variety of 'morality plays'. Besides the original 2007 study, which featured a character attempting to climb up the hill and being pushed up or down, respectively, by two other characters ('hill' paradigm), infants' preference for prosocial agents has been explored in a number of instrumental helping scenarios: one requiring a box to be opened to retrieve a desired object ('box' paradigm [3]); another requiring a ball to be returned to its owner, who accidentally dropped it while playing ('ball' paradigm [3]); others requiring a shelf to be knocked or a door to be opened to make an out-of-reach object accessible [4,5].

Adding to the situational breadth of early social evaluation that these studies attested, others showed this to be a nuanced and sophisticated phenomenon. Already in their first year of life, infants appear sensitive to epistemic states and overt intentions: they prefer intentional over accidental helpers, but accidental over intentional hinderers [4], and unsuccessful helpers over unsuccessful hinderers [5]. Additionally, infants show a preference for helpers only when these know the particular goal the helpee is trying to accomplish [6,7]. Moreover, infants do not choose characters on the basis of the mere valence of the actions they performed, but interpret them in context, preferring a character who 'punishes'—i.e. acts antisocially towards—a previous hinderer over a character who helps her [8,9].

Beyond instrumental helping, a preference for prosocial characters has been found in a number of other sociomoral domains. In the domain of physical aggression, for instance, infants preferred victims over perpetrators [10,11], and third-party characters intervening in a conflict to shield victims from ongoing aggression over passive bystanders [12]. Similarly, in the domain of resource allocation, infants have been repeatedly shown to prefer fair distributors over unfair ones [13–15]. Modified versions of the manual choice paradigm have also been recently used to investigate whether similar evaluative tendencies exist in non-human animals, such as bonobos [16], capuchin monkeys [17] and dogs [18].

Despite the recent growth of the literature on early sociomoral evaluation, attempts to replicate the findings by Hamlin *et al.* have yielded mixed results. For example, using the original 'hill scenario', Cowell & Decety [19] found no significant preference for helpers in 12- to 24-month-olds (see also Colaizzi [20]). Similarly, Scarf *et al.* [21] suggested that low-level perceptual features, rather than inferred sociomoral dispositions, could adequately explain infants' preference for prosocial characters (though see Hamlin [6] for a rebuttal of this claim). Using the 'box scenario', Salvadori *et al.* [22] found no preference for helpers across two experimental attempts. A similar lack of preference was documented by Nighbor *et al.* [23] with 5- to 16-month-olds. Conversely, using the 'ball scenario', Scola *et al.* [24] reported a significant preference for prosocial characters in 12- to 24- and 24- to 36-month-olds, whereas Shimizu *et al.* [25] documented a similar, albeit weaker, preference in 15- to 18-month-olds, but not in younger age groups. It is worth noting, however, that previous replication attempts have followed the methods of the original studies to varying degrees of fidelity. Differences in stimuli materials and procedural details might have conceivably affected infants' responses.

In a recent meta-analysis, Margoni & Surian [26] reviewed 26 published and unpublished studies using manual choice measures to investigate early sociomoral evaluation. While their analysis revealed an overall significant tendency to prefer prosocial characters across studies, the authors cautioned about the possibility of publication bias and the under-reporting of negative findings (file drawer problem). Importantly, Margoni and Surian also attested the presence of a laboratory effect: research conducted by Hamlin and collaborators tends to generate larger effect sizes compared to studies done by independent laboratories. On these grounds, the authors called for more and sufficiently powered replications.

Here, we conducted a conceptual replication of the original study by Hamlin *et al.* [1]. Our study differs from the original in three potentially important ways. Firstly, we tested 15-month-old infants, an age group slightly older than the infants tested in similar studies. While Margoni & Surian's meta-analysis [26] found no significant effect of age on infants' preference for prosocial characters, the participants' mean age in the studies reviewed was approximately 13 months (390 days). Secondly, we did not present the stimuli in the form of a live puppet show, but as video animations on a screen, which were generously provided to us by Woo and Hamlin. Although Margoni & Surian [26] found no effect of presentation mode (live versus video), a majority of the studies in their sample were based on live puppet shows. Thirdly, instead of using a habituation procedure, we employed a familiarization procedure, presenting the stimuli for a fixed amount of time across infants. This was aimed at mitigating the problem of fussiness and high drop-out rates, common with older infants when using habituation designs.

Crucially, these modifications were implemented under recommendation of Woo and Hamlin, who used the same video stimuli and familiarization procedure for their own in-laboratory replication (in preparation) of the original Hamlin *et al.* study [1]. In said replication, Woo and Hamlin found a significant preference for the helper character in a sample of 32 infants (23 of 32; reported in Margoni & Surian's meta-analysis [26]).

# 2. Methods

This article received results-blind in-principle acceptance (IPA) at Royal Society Open Science. Following IPA, the accepted Stage 1 version of the manuscript, not including results and discussion, was preregistered on the OSF (https://osf.io/krms8). The preregistration was produced after data collection and analysis.

## 2.1. Piloting phase

Before testing our experimental sample, we conducted a pilot with 24 infants aged 14–16 months. During the piloting phase, we sent video recordings of the participants to Hamlin (written permission for data sharing was obtained from the parents), who kindly provided helpful feedback on the procedure, and we subsequently implemented her suggestions. Testing of the experimental sample began only after Hamlin had confirmed that our procedure was sufficiently close to the original.

## 2.2. Participants

Thirty-two 14- to 16-month-old infants participated in the study (mean age: 15.18 months, range: 431–492 days, 20 males). The sample size was determined prior to data collection and was twice the sample of 10-month-olds and more than twice the sample of 6-month-olds tested in Hamlin *et al*. [1]. An additional 19 infants were tested but not included in the final sample due to failing to produce a choice at test ($n = 7$), inattentiveness during familiarization ($n = 5$), fussiness ($n = 4$), experimenter error ($n = 2$) and technical failure ($n = 1$). Participants were full-term infants with no reported health or developmental issues. Infants were recruited from the database of the Cognitive Development Center, which includes contact information of parents volunteering to participate in research with their children. Data collection took place between January and May 2018.

## 2.3. Ethical approval

Carers were informed about the nature and possible consequences of the study, and gave informed consent for their child to participate. We obtained ethical approval for this research from the United Ethical Review Committee for Research in Psychology (EPKEB) in Hungary, and it was conducted according to the ethical rules and standards regarding psychological experimentation in Hungary.

## 2.4. Materials and apparatus

During the familiarization phase, infants were seated in their carer's lap in a dimly lit room, approximately 60 cm away from a TV screen of 100 cm diagonal size. The stimuli were generated by Woo and Hamlin using Blender animation software (https://www.blender.org/download), and were presented on a screen using PsyScope X [27] controlled by a Mac Mini computer.

The objects for the manual choice procedure were printed-out versions of the blue square and yellow triangle characters from the stimuli videos (square: $13 \times 13$ cm, triangle: $15.5 \times 13.5$ cm). The printout graphics were converted from RGB to CMYK colour space and adjusted, so that the colour of the printed characters matched those on screen as closely as possible. Printouts were glued onto figures made of stacked cardboard, to mimic the three-dimensional appearance of the characters in the video. The figures were then wrapped with a transparent plastic cover, to protect them from wear. The figures were attached with removable adhesive putty onto a white board ($50 \times 36$ cm) at a distance of 19 cm from each other, 3 cm from the sides of the board and 3 cm from the bottom of the board.

During the familiarization, Experimenter 1, who ran the study and coded the infants' looking behaviour online, was seated in the same room as the child, hiding behind a black curtain. Experimenter 2, who performed the manual choice task, also hid behind the curtain during the familiarization phase. To ensure that Experimenter 2 was blind to condition, she had no visual access to the screen displaying the stimuli.

## 2.5. Procedure

Before the familiarization phase, Experimenter 2 briefed the carer on how to position herself for the manual choice task. The carer was instructed to turn her chair away from the screen, place her feet

behind a tape marking on the floor and have the child sit on her knees while supporting the child by the ribcage. After this training on the choice phase, the carer was asked to turn back towards the screen for the familiarization phase and to keep her eyes closed for the whole duration of the study.

*Familiarization phase.* Infants watched a total of six familiarization trials featuring three helping and three hindering events, alternated. Each trial was preceded by a brief attention-getter (a flashing checkerboard accompanied by the sound of a xylophone slide) which played until the child gazed back at the screen. The two familiarization events were matched in timing and overall duration (17 s).

Both events took place on a light-blue sky background and a dark green hill, extending from the bottom left to the top right corner of the screen. The hill plateaued halfway and at the top.

Each event started with a character (a small red circle with eyes pointing to the top of the hill; hereinafter, Protagonist) located at the bottom of the hill. After a bell sound, the Protagonist moved to the intermediate plateau and bounced up and down twice with her eyes directed towards the viewer (2 s). She then attempted to climb the top plateau twice, each time reaching up to two-thirds of the steep incline and sliding back down to the intermediate plateau (8 s). At this point, the helper or hinderer appeared, again to the sound of a bell (helper: from the bottom left of the screen; hinderer: from the top right of the screen). As the Protagonist attempted to climb the steep incline to the top plateau for a third time, the helper/hinderer (whose eyes were directed to the top or bottom of the hill, respectively) moved towards the Protagonist and, with two repeated shoves (accompanied by a knocking sound), pushed the Protagonist up to the top plateau or down to the bottom one (4 s). Upon reaching either the bottom or the top of the hill, the Protagonist came to a standstill, while the other character exited the scene from the location where she initially appeared (3 s).

Each trial ended with a still frame, kept on screen until the infants had looked away for a minimum of two consecutive seconds or until 30 s had elapsed.

*Test phase.* Immediately after the end of the familiarization phase, the screen turned black and a soft guitar tune started playing (also provided by Woo and Hamlin). Experimenter 2, following a cue from Experimenter 1, entered the testing room from behind the curtain, turned on the light and instructed the carer to assume the previously practised position for the manual choice task and to close her eyes again afterwards. Experimenter 2 kneeled down in front of the child and addressed her while making eye contact: 'Szia [name of child]! Kivel szeretnél játszani?', which translates to 'Hi [name of child]! Who would you like to play with?'. Then, she lowered her gaze to the chin of the child and flipped over the board with the two characters. The board was moved towards the infant and turned slightly downward at approximately a 30° angle, so that the figures were within the infant's reach but required participants to stretch out their arms in order to touch them. After the board had been flipped over, the experimenter made sure not to pull the board away while the infant was reaching out for a character, as this might convey to the infant that her intended choice was 'wrong' (JK Hamlin 2017, personal communication).

If the infant did not produce any visually guided reaching after approximately 30 s, Experimenter 2 verbally encouraged the infant by saying, for instance, 'Figyelj!' (Pay attention!), 'Nézd meg őket!' (Look at them!), or 'Bátran!' (Be brave!), and repeating the original question. If no choice was produced after 2 min, the experiment was terminated.

The following factors were counterbalanced in the study: the identity of helper and hinderer during familiarization (blue square versus yellow triangle), the order of event presentation (helping first versus second) and the position of the characters on the board (helper on the right versus left side). The condition that each infant was assigned to was randomly selected before testing.

## 2.6. Coding and analyses

The dependent variable was the infants' choice of the helper or hinderer character, assessed by their reaching to one of the figures on the board. In order to be counted as a choice, the reaching had to be visually guided: i.e. infants had to look at a character before and while touching it. If infants reached for a figure while looking elsewhere, they were given the opportunity to produce another reach within the 2 min time window. If infants touched both figures, but looked only at one prior to establishing contact, this was coded as a choice for the figure they looked at.

Experimenter 2 judged online whether the infant had reached unambiguously for one of the figures and thus whether to terminate a trial. Choices were coded offline from the videos by Experimenter 1, and recoded by an independent second coder blind to the experimental condition, reaching 93.75% of agreement. Two infants judged by the second coder to have made no clear choice were removed from the final sample and replaced.

In order to be included in the final data analysis, infants had to watch at least 50% of the duration of each helping/hindering event (from the onset of physical contact between the protagonist and the helper/hinderer to the end of the pushing action) in all six trials. This stringent criterion of attentiveness was meant to ensure that each infant attended to the crucial social interactions differentiating helper and hinderer for a sufficient number of times. Including the manual choice data from the infants who did not meet this criterion did not affect the results.

In order to assess whether infants showed preference for the helper character, we performed a one-tailed binomial test on the number of infants who chose the helper and the total number of infants included in the analysis against the probability of 0.5 as chance level (as was done by Hamlin *et al.* [1]). Statistical analyses were performed with R, the lme4 package [28] and the BayesFactor package [29]. Data are available at https://osf.io/kh5r4/.

# 3. Results

## 3.1. Hypothesis-driven analyses

Sixteen out of 32 children directed their first visually guided reach to the helper (one-tailed binomial test $p = 0.57$; 95% CI: 0.344–1.0). Thus, infants did not display a preference for either the helper or the hinderer character. When including in the analysis the 5 infants who were excluded due to inattentiveness during the familiarization phase, 20 out of 37 reached for the helper (one-tailed binomial test $p = 0.371$; 95% CI: 0.394–1.0).

## 3.2. Further results

In a Bayesian analysis with a null model of $p = 0.5$ and an alternative model with a uniform prior (implemented in the BayesFactor package by an 'ultrawide' scale parameter of 1), the data from our study yielded a Bayes factor of 4.618 in favour of $H_0$, indicating moderate support for the null hypothesis of no effect [29].

Infants' choice was not significantly influenced by their gender (9 of 20 male infants chose the helper, while 7 of 12 females did), characters' features (20 of 32 infants reached for the yellow triangle), characters' location on the board (17 infants reached for the figure on the right) and order of familiarization events (12 infants reached for the agent they last saw).

During the manual choice, a subset of infants did not unambiguously direct their gaze at both characters before producing a choice. This, however, did not affect the results: 12 of 24 of those infants who looked at both characters chose the helper, whereas 4 of 8 of those who only looked at one character reached for the helper.

Since we presented infants with a fixed number of trials in a familiarization design, the present failure may also be due to insufficient exposure to the two characters' actions. Indeed, infants' average looking times from the first three trials (12.81 s) to the last three trials (9.99 s) decreased by 22%, thus failing to meet the habituation criterion previously adopted by Hamlin (i.e. decrease in looking by 50% from the first three to the last three trials). To assess the effects that the overall weak level of habituation had on infants' choices, we examined whether stronger habituation predicted a higher likelihood of reaching for the helper, but found no support for this hypothesis ($\beta = -0.002$, s.e. $\beta = 0.007$, $p = 0.816$, logistic regression model).

We also analysed whether the amount of looking to the two types of familiarization events may have influenced the infants' behaviour at test. In line with previous studies, we found no difference in the mean looking times to the still frames following the two events (helping: 11.41 s; hindering: 11.39 s). We fit a mixed-effects linear regression model predicting log looking time from familiarization event type with a subject-random intercept. Model comparison revealed no significant looking time difference between the event types ($\chi^2 = 0.03$, $p = 0.864$). Moreover, infants did not tend to choose the agent they attended to longer on average during familiarization (16 of 32 reached for the character they had looked at longer).

# 4. Discussion

In the present study, we attempted to replicate Hamlin *et al.*'s [1] finding that infants preferentially reach for helpful over hindering characters [1]. In that study, 92.9% of infants exhibited such preferences (14 of 16 10-

month-olds and 12 of 12 6-month-olds). By contrast, only 50% of infants did so in our study (16 of 32). Therefore, we could not reproduce the original findings. There are several potential explanations for such a failure. Our study differed from the original in three potentially relevant ways: firstly, we tested infants from an older age group (15-month-olds, rather than 6- and 10-month-olds); secondly, we used three-dimensional animated videos rather than a live puppet show to expose infants to the helping and hindering events; and thirdly, we used a familiarization rather than a habituation design.

Any of these deviations from the original study may have potentially contributed to our results. For instance, it is conceivable that six familiarization trials were insufficient for infants to learn about the agents' respective dispositions. Supporting this possibility, the average decrease in looking times during familiarization was insufficient to reach the habituation criterion adopted by Hamlin in previous studies (decrease of 50% from the first three to the last three trials). It should be noted, however, that prior studies [7,12,14] and the in-laboratory replication onto which our study was modelled successfully elicited a preference for prosocial characters by means of familiarization.

Alternatively, infants may have had troubles mapping the cardboard replicas of helper and hinderer to the three-dimensional animated characters they were familiarized to. While this remains a genuine possibility, several studies reported preferential reaching for replicas of prosocial characters presented on screen [12,14,24,30], and the meta-analysis found no effect of presentation type on infants' preferences [26].

Our study used animations and familiarization following recommendations by Woo and Hamlin, who found these stimuli and design to be suitable for eliciting social evaluation in infants older than 12 months of age. It should be noted, however, that the percentage of infants reaching for the helper in their in-laboratory replication was lower than in the original study [1], and failed to show the effect in two additional samples of younger infants (8-month-olds: 21/32; 10-month-olds: 15/32; as reported in the supplementary materials of Margoni & Surian [26]). These differences raise the possibility that familiarizing infants to animations may not be as effective in eliciting social evaluation as habituating them to live-action puppet shows.

It is also possible that other unforeseen methodological differences, some of which may be hard or impossible to control for, contributed to our failed replication. Such differences may concern, for instance, the physical set-up of the testing environment, the cultural background of the population tested or, more likely, the behaviour of the experimenters involved in the study. On this note, it is, however, worth noting that, unlike other replication attempts, ours benefited from the close and careful scrutiny of the experimenters' behaviour by Hamlin herself. Her feedbacks during the piloting phase allowed us to fine-tune the procedure of character presentation in ways that other studies could not.

Finally, current evidence suggests that the underlying effect size of infants' preference for helpful characters may be smaller than originally assumed. The meta-analysis by Margoni & Surian [26] estimated that on average 64% of infants in the studies reviewed reached for the prosocial character. Importantly, however, the strength of infants' preference was affected by the sociomoral domain tested: 77% of infants preferred the prosocial character after observing giving versus taking events, 69% after observing fair versus unfair distributions and only 63% after observing helping versus hindering events. Although instrumental helping represented the domain with the lowest percentage of infants' choice of the prosocial agent, this was nevertheless considerably higher than the percentage (50%) obtained in our study.

In a recent paper, Margoni & Shepperd [31] argued that individual replication studies ought not to be considered as confirming or disconfirming an effect, but rather should be pooled together to produce a better estimate of the true underlying effect size of the phenomenon at hand. If original studies are underpowered, as is often the case in infant research, replications with a relatively wide range of results may technically be taken as confirming the original finding if they fall within a 'prediction interval' of potential outcomes. This said, our proportion of 50% helper choices falls outside the value range (0.59–1.0) defined by the 95% prediction interval proposed by Margoni & Shepperd for a replication of Hamlin et al.'s [1] study with $n = 32$, and thus cannot be considered confirmatory.

The present replication sheds further light on the robustness of the phenomenon of early sociomoral evaluation and the conditions under which it can be reliably elicited. It also contributes to broader methodological debates on the replicability of findings in developmental science, and reaffirms the need, already voiced by Margoni & Surian, for multi-laboratory replication initiatives [32] that could adequately assess the influence of potentially mediating factors.

Ethics. We obtained ethical approval for this research from the United Ethical Review Committee for Research in Psychology (EPKEB) in Hungary, and it was conducted according to the ethical rules and standards regarding psychological experimentation in Hungary.

Data accessibility. The dataset generated and analysed during the current study is available at the OSF repository (https://osf.io/kh5r4/), as are the stimuli (shared with permission from Brandon Woo and Kiley Hamlin).

Authors' contributions. L.S. performed research and analysed the data; L.S., G.C. and D.T. wrote the paper.

Competing interests. The authors declare no competing interests.

Funding. This research has received partial funding from the European Research Council (ERC) under the European Union's Horizon 2020 research and innovation programme under grant agreement no. 742231 (PARTNERS).

Acknowledgements. We thank K. Hamlin and B. Woo for help with implementing the experimental procedure and providing the stimuli, M. Nagy for assistance with data collection and P. Rácz for statistical advice.

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
