## [Reviewer comments · Royal Society Open Science]

Review History

Decision letter (RSOS-191747.R0)

11-Oct-2019

Dear Ms Schlingloff,

I write you in regards to manuscript RSOS-191747 entitled "Do 15-month-old infants prefer helpers? A replication of Hamlin *et al.* (2007)" which you submitted to Royal Society Open Science.

We routinely triage submissions for adherence to the Replication Policy guidelines. For submissions that have promise but are not yet suitable for in-depth Stage 1 review, we offer feedback to help authors maximise the chances that reviewers will respond positively to a resubmission.

We have concluded that your submission is not yet suitable for in-depth review and is therefore rejected at this time, but we believe it will be suitable once several issues are addressed. We therefore invite a resubmission. Further comments from the Associate Editor may be found at the end of this letter.

If you wish to revise your manuscript in light of the below comments please submit your manuscript as a new submission and mention this previous manuscript ID in your covering letter. You should also provide a detailed response to the below comments in the cover letter. Author guidelines may be found at <https://royalsocietypublishing.org/rsos/replication-studies#AuthorsGuidance>.

Thank you for considering Royal Society Open Science for the publication of your Replication study.

Kind regards,
Anita Kristiansen
Editorial Coordinator
Royal Society Open Science
openscience@royalsociety.org

on behalf of Professor Chris Chambers
openscience@royalsociety.org

Associate Editor Comments to Author:
Associate Editor

Comments to the Author:

In order to proceed to in-depth review, please expand the Coding and Analyses section to make clear the exact analyses that will be reported in the (eventual) Results. Please do so without anticipating the actual results themselves. The description of the statistical analyses must be sufficiently detailed for reviewers to be able to assess their validity and similarity to the analyses in the original target study.

Author's Response to Decision Letter for (RSOS-191747.R0)

See Appendix A.

RSOS-191795.R0

Review form: Reviewer 1 (Ashley Thomas)

Do you have any ethical concerns with this paper?

Yes

Have you any concerns about statistical analyses in this paper?

Yes

Recommendation?

Accept with minor revision

Comments to the Author(s)

I support an 'in principle acceptance' of this publication. My suggestions are (1) to justify the sample size if it was based on the sample size in Woo & Hamlin, then say so). (2) The way it is written now is that the interpretations of the original authors are the only interpretation of a reaching preference for helpers. This work is undoubtedly useful for the field. However, if the

authors find the same effect, it does not necessarily mean that the original interpretation of the data is the only plausible one.

Review form: Reviewer 2

Do you have any ethical concerns with this paper?

No

Have you any concerns about statistical analyses in this paper?

No

Recommendation?

Accept with minor revision

Comments to the Author(s)

This is an excellent proposal. I am firmly in support of this project; the effect being studied is of central importance to theories in social cognitive development, the record in the literature is deeply confusing, and the authors' approach of preregistering their methods while collaborating extensively with the original authors to establish an ideal but unbiased protocol is the best way to go about executing a replication. Bravo to all involved. I have only a few small suggestions, none of which need be followed for me to support accepting this manuscript.

Can the authors provide a power analysis outlining their expectations about effect size and justifying intended sample size? Going off the previous results w/ the same stimulus set (23/32), what are odds of observing effect in new sample of 32? For high certainty of replicating a true effect (which seems like the goal here), 2.5x original sample has sometimes been recommended (Simonsohn, 2015)... the preregistered one-tailed test does help, but I would suggest at least extending the sample to 1.5x the original size of Woo & Hamlin (N = 48), so that the outcome of the study is more convincing to readers who come in with preconceived notions about the status of this effect from either direction.

The description of parent instructions in the first paragraph of Procedure section is a little hard to follow; I initially interpreted it as parents were instructed to have their chair turned away from the screen but their head/body facing the screen throughout the study. Instead, I think the authors meant that after training on the choice phase, parents were asked to turn their chair back toward the screen for the familiarization phase. Wording along these lines might prevent my (possibly silly) misinterpretation.

What if an infant reaches out and touches one character while looking elsewhere, e.g. at the other character or the experimenter? Is that infant excluded? Are any attempts made to discourage this and elicit a valid reach, e.g. by pulling board back if a baby is reaching without looking, and encouraging the babies to look at the characters before extending the board again?

How will coder reliability be assessed? Does Exp 2 also judge and record the infant's response after administering the choice procedure?

The familiarization looking inclusion criterion seems strict, especially for this age range. What if an infant extensively habituates after substantial looking over the first 4 trials? Is this criterion following Hamlin & Woo? If not, I would encourage relaxing criterion to 50% of trial for at least 2 familiarization trials of each type, rather than all three. This better matches what we would ask of an infant in a habituation procedure, and it has already been noted that this age range is often too impatient for those.

Decision letter (RSOS-191795.R0)

20-Dec-2019

Dear Ms Schlingloff

On behalf of the Editors, I am pleased to inform you that your Stage 1 Replication RSOS-191795 entitled "Do 15-month-old infants prefer helpers? A replication of Hamlin et al. (2007)" has been accepted in principle for publication in Royal Society Open Science subject to minor revision in accordance with the referee and editor suggestions. Please find their comments at the end of this email.

The reviewers and handling editors have recommended publication, but also suggest some minor revisions to your manuscript. Therefore, I invite you to respond to the comments and revise your manuscript.

Please you submit the revised version of your manuscript within 30 days (i.e. by the 28-Dec-2019). If you do not think you will be able to meet this date please let me know immediately.

Full author guidelines can be found here <https://royalsocietypublishing.org/rsos/replication-studies#AuthorsGuidance>.

Kind regards,
Professor Chris Chambers
Royal Society Open Science
openscience@royalsociety.org

on behalf of Professor Chris Chambers (Registered Reports Editor, Royal Society Open Science)
openscience@royalsociety.org

Associate Editor Comments to Author (Professor Chris Chambers):

Associate Editor: 1

Comments to the Author:

Two reviewers have now assessed the Stage 1 submission. Both are very positive and recommend in-principle acceptance following minor revisions to improve clarity and provide additional methodological detail and justification of design decisions. Both reviewers note the importance of justifying the chosen sample size, so please pay particular attention to this issue in revision.

Reviewers' comments to Author:

Reviewer: 1

Comments to the Author(s)

I support an 'in principle acceptance' of this publication. My suggestions are (1) to justify the sample size if it was based on the sample size in Woo & Hamlin, then say so). (2) The way it is written now is that the interpretations of the original authors are the only interpretation of a reaching preference for helpers. This work is undoubtedly useful for the field. However, if the authors find the same effect, it does not necessarily mean that the original interpretation of the data is the only plausible one.

Reviewer: 2

Comments to the Author(s)

This is an excellent proposal. I am firmly in support of this project; the effect being studied is of central importance to theories in social cognitive development, the record in the literature is deeply confusing, and the authors' approach of preregistering their methods while collaborating extensively with the original authors to establish an ideal but unbiased protocol is the best way to go about executing a replication. Bravo to all involved. I have only a few small suggestions, none of which need be followed for me to support accepting this manuscript.

Can the authors provide a power analysis outlining their expectations about effect size and justifying intended sample size? Going off the previous results w/ the same stimulus set (23/32), what are odds of observing effect in new sample of 32? For high certainty of replicating a true effect (which seems like the goal here), 2.5x original sample has sometimes been recommended (Simonsohn, 2015)... the preregistered one-tailed test does help, but I would suggest at least extending the sample to 1.5x the original size of Woo & Hamlin (N = 48), so that the outcome of the study is more convincing to readers who come in with preconceived notions about the status of this effect from either direction.

The description of parent instructions in the first paragraph of Procedure section is a little hard to follow; I initially interpreted it as parents were instructed to have their chair turned away from the screen but their head/body facing the screen throughout the study. Instead, I think the authors meant that after training on the choice phase, parents were asked to turn their chair back toward the screen for the familiarization phase. Wording along these lines might prevent my (possibly silly) misinterpretation.

What if an infant reaches out and touches one character while looking elsewhere, e.g. at the other character or the experimenter? Is that infant excluded? Are any attempts made to discourage this and elicit a valid reach, e.g. by pulling board back if a baby is reaching without looking, and encouraging the babies to look at the characters before extending the board again?

How will coder reliability be assessed? Does Exp 2 also judge and record the infant's response after administering the choice procedure?

The familiarization looking inclusion criterion seems strict, especially for this age range. What if an infant extensively habituates after substantial looking over the first 4 trials? Is this criterion following Hamlin & Woo? If not, I would encourage relaxing criterion to 50% of trial for at least 2 familiarization trials of each type, rather than all three. This better matches what we would ask of an infant in a habituation procedure, and it has already been noted that this age range is often too impatient for those.

Author's Response to Decision Letter for (RSOS-191795.R0)

See Appendix B.

Decision letter (RSOS-191795.R1)

21-Jan-2020

Dear Ms Schlingloff

On behalf of the Editor, I am pleased to inform you that your Manuscript RSOS-191795.R1 entitled "Do 15-month-old infants prefer helpers? A replication of Hamlin et al. (2007)" has been accepted in principle for publication in Royal Society Open Science. The reviewers' and editors' comments are included at the end of this email.

You may now progress to Stage 2 and complete the study as approved.

Please note that you must now register your approved protocol on the Open Science Framework (<https://osf.io/tr>), using the 'Submit your approved Registered Report' option and then the 'Registered Report Protocol Preregistration' option. Please use the Registered Report option even though your article is being accepted as a Stage 1 Replication. Further into the registration process, in the Journal Title field enter 'Royal Society Open Science (Replication article type, Results-Blind track)'. Please note that a time-stamped, independent registration of the protocol is mandatory under journal policy, and manuscripts that do not conform to this requirement cannot be considered at Stage 2. The protocol should be registered unchanged from its current approved state. Please include a URL to the protocol in your Stage 2 manuscript, and because you submitted via the Results-Blind track please note in the manuscript that the pre-registration was performed after data analysis (e.g. 'This article received results-blind in-principle acceptance (IPA) at Royal Society Open Science. Following IPA, the accepted Stage 1 version of the manuscript, not including results and discussion, was preregistered on the OSF (URL). This preregistration was performed after data analysis.')

We would be grateful if you could now update the journal office as to the anticipated completion date of your study.

Following completion of your study, we invite you to resubmit your paper for peer review as a Stage 2 Replication. Please note that your manuscript can still be rejected for publication at Stage 2 if the Editors consider any of the following conditions to be met:

- The Introduction and methods deviated from the approved Stage 1 submission (required).
- The authors' conclusions were not considered justified given the data.

We encourage you to read the complete guidelines for authors concerning Stage 2 submissions at: <https://royalsocietypublishing.org/rsos/replication-studies#AuthorsGuidance>. Please especially note the requirements for data sharing and that withdrawing your manuscript will result in publication of a Withdrawn Registration.

Once again, thank you for submitting your manuscript to Royal Society Open Science and I look forward to receiving your Stage 2 submission. If you have any questions at all, please do not hesitate to get in touch. We look forward to hearing from you shortly with the anticipated submission date for your stage two manuscript.

Kind regards,
Professor Chris Chambers
Royal Society Open Science
openscience@royalsociety.org

on behalf of Professor Chris Chambers (Registered Reports Editor, Royal Society Open Science)
openscience@royalsociety.org

Author's Response to Decision Letter for (RSOS-191795.R1)

See Appendix C.

RSOS-191795.R2 (Revision)

Review form: Reviewer 1 (Ashley Thomas)

Is the manuscript scientifically sound in its present form?

Yes

Do you have any ethical concerns with this paper?

No

Have you any concerns about statistical analyses in this paper?

No

Recommendation?

Accept as is

Comments to the Author(s)

Thomas & Sarnecka presented live puppet shows to infants, not animations.

You could cite Powell, Spelke, 2018 (third party preferences for imitators) who I believe used a video with manual choice; there are, I believe others by Woo et al.

Review form: Reviewer 2

Is the manuscript scientifically sound in its present form?

Yes

Do you have any ethical concerns with this paper?

No

Have you any concerns about statistical analyses in this paper?

No

Recommendation?

Accept with minor revision

Comments to the Author(s)

In general, the study and results reported here conform to the methods and analysis plans described in Stage 1, and I am happy to recommend the paper for publication.

The authors include a few exploratory analyses of factors that may have affected infants' helper preferences. One they did not include but that may be of interest given the overall weak level of habituation across the six familiarization is to ask if stronger habituation (i.e. greater proportional

decrease in looking from the first three trials to the last three) predicted a higher likelihood of reaching for the helper.

I do also have one concern/disagreement with the language in the discussion section. The authors refer to the unpublished Woo & Hamlin data repeatedly to argue that there was no reason to suspect their changes from the Hamlin et al 2007 procedure would affect infants' preference. This already seemed a bit unfair to me, going off the information in the manuscript alone, which reports a helper preference percentage in the Woo & Hamlin data (71%) that's notably lower than the one in the pooled Hamlin et al 2007 data (>90%). I went to the Margoni & Surian meta analysis to check whether Woo & Hamlin also tested an age range similar to the authors of the current ms, to consider recommending that the authors note that the difference in effect size between Hamlin et al 2007 and Woo & Hamlin may have been real (though not large enough overall N to tell) and also affecting the current sample, and made a discovery: the sample in which 23/32 infants reached for the helper was only one of three samples Woo & Hamlin report collecting using the animated stimuli. Those infants were 12 months old, but Woo & Hamlin also collected 8-month-old and 10-month-old samples in which 21/32 and 15/32 infants, respectively, reached for the helper (see Margoni & Surian, supplemental materials). Neither of those proportions meet the (standard, two-tailed .05) criterion for statistical significance. Moreover, if you pool the Hamlin et al 2007 data and compare to pooled Woo & Hamlin data, the preference rates for the two stimulus types are reliably different.

So, in 2 of 3 samples, all within the age range of the original live-action preference effects from the 2007 paper, the original author (Hamlin) failed to find reliable evidence for a helper preference with the stimuli (and familiarization procedure? unclear) used here. It thus seems a bit unfair to claim that there's "no reason to suspect in advance that these factors would affect infants' behaviors," while mentioning only the one sample that did replicate the effect, especially given that it was somewhat underpowered to detect an effect of the observed size (71% preference for helper).

The timelines of research projects and publications are often long, and the authors may well have begun their project before the Margoni & Surian meta analysis was published. Likewise, Woo & Hamlin may have only disclosed the positive 12-month-old results, so the authors may not have known about the null results with 8- and 10-month-old infants. If that's the case, it's fine for them to say that they selected their procedure based on the evidence/recommendation they had at the time of study design. However, I would recommend including stronger emphasis on (or at least some mention of) the within-lab discrepancy between results of live-action and animated stimuli obtained by the original authors. They could also enumerate other potential reasons, beyond screen-to-object mapping, such a discrepancy would exist (e.g. animated stimuli fail to elicit same goal attribution to protagonist, especially in 2D environment where there may be no prior that moving upward is difficult? necessarily non-blind puppeteers have unintended experimenter effects on preferences?).

Decision letter (RSOS-191795.R2)

26-Feb-2020

Dear Ms Schlingloff

On behalf of the Editor, I am pleased to inform you that your Stage 2 Replication submission RSOS-191795.R2 entitled "Do 15-month-old infants prefer helpers? A replication of Hamlin et al. (2007)" has been accepted for publication in Royal Society Open Science subject to minor revision in accordance with the referee suggestions. Please find the referees' comments at the end of this email.

The reviewers and Subject Editor have recommended publication, but also suggest some minor revisions to your manuscript. Therefore, I invite you to respond to the comments and revise your manuscript.

Please also ensure that all the below editorial sections are included where appropriate (a non-exhaustive example is included in an attachment):

- Ethics statement

- Data accessibility

If you wish to submit your supporting data or code to Dryad (<http://datadryad.org/>), or modify your current submission to dryad, please use the following link:
<http://datadryad.org/submit?journalID=RSOS&manu=RSOS-191795.R2>

- Competing interests

- Authors' contributions

- Acknowledgements

- Funding statement

Because the schedule for publication is very tight, it is a condition of publication that you submit the revised version of your manuscript within 7 days (i.e. by the 05-Mar-2020). If you do not think you will be able to meet this date please let me know immediately.

- 1) A text file of the manuscript (tex, txt, rtf, docx or doc), references, tables (including captions) and figure captions. Do not upload a PDF as your "Main Document".
- 2) A separate electronic file of each figure (EPS or print-quality PDF preferred (either format should be produced directly from original creation package), or original software format)
- 3) Included a 100 word media summary of your paper when requested at submission. Please ensure you have entered correct contact details (email, institution and telephone) in your user account
- 4) Included the raw data to support the claims made in your paper. You can either include your data as electronic supplementary material or upload to a repository and include the relevant DOI within your manuscript
- 5) Included your supplementary files in a format you are happy with (no line numbers, Vancouver referencing, track changes removed etc) as these files will NOT be edited in production

Kind regards,
Andrew Dunn
Royal Society Open Science
openscience@royalsociety.org

Associate Editor Comments to Author (Professor Chris Chambers):

The two reviewers who assessed the Stage 1 manuscript have now reviewed the Stage 2 submission. Both are broadly satisfied with the manuscript, suggesting minor revisions including a potential additional analysis and moderation of the Discussion. In revising the manuscript, please ensure that no changes are made to the approved Stage 1 part of the manuscript aside from correcting any errors of fact. Any additional literature or rationale should instead be covered in the Discussion. Provided the authors respond thoroughly to the issues raised, full acceptance should be forthcoming without requiring further in-depth review.

Reviewer: 1

Comments to the Author(s)

Thomas & Sarnecka presented live puppet shows to infants, not animations.

You could cite Powell, Spelke, 2018 (third party preferences for imitators) who I believe used a video with manual choice; there are, I believe others by Woo et al.

Reviewer 2:

Comments to the Author(s)

In general, the study and results reported here conform to the methods and analysis plans described in Stage 1, and I am happy to recommend the paper for publication.

The authors include a few exploratory analyses of factors that may have affected infants' helper preferences. One they did not include but that may be of interest given the overall weak level of habituation across the six familiarization is to ask if stronger habituation (i.e. greater proportional decrease in looking from the first three trials to the last three) predicted a higher likelihood of reaching for the helper.

I do also have one concern/disagreement with the language in the discussion section. The authors refer to the unpublished Woo & Hamlin data repeatedly to argue that there was no reason to suspect their changes from the Hamlin et al 2007 procedure would affect infants' preference. This already seemed a bit unfair to me, going off the information in the manuscript alone, which reports a helper preference percentage in the Woo & Hamlin data (71%) that's notably lower than the one in the pooled Hamlin et al 2007 data (>90%). I went to the Margoni & Surian meta analysis to check whether Woo & Hamlin also tested an age range similar to the authors of the current ms, to consider recommending that the authors note that the difference in effect size between Hamlin et al 2007 and Woo & Hamlin may have been real (though not large enough overall N to tell) and also affecting the current sample, and made a discovery: the sample in which 23/32 infants reached for the helper was only one of three samples Woo & Hamlin report collecting using the animated stimuli. Those infants were 12 months old, but Woo & Hamlin also collected 8-month-old and 10-month-old samples in which 21/32 and 15/32 infants, respectively, reached for the helper (see Margoni & Surian, supplemental materials). Neither of those proportions meet the (standard, two-tailed .05) criterion for statistical significance. Moreover, if you pool the Hamlin et al 2007 data and compare to pooled Woo & Hamlin data, the preference rates for the two stimulus types are reliably different.

So, in 2 of 3 samples, all within the age range of the original live-action preference effects from the 2007 paper, the original author (Hamlin) failed to find reliable evidence for a helper preference with the stimuli (and familiarization procedure? unclear) used here. It thus seems a bit unfair to claim that there's "no reason to suspect in advance that these factors would affect infants' behaviors," while mentioning only the one sample that did replicate the effect, especially given that it was somewhat underpowered to detect an effect of the observed size (71% preference for helper).

The timelines of research projects and publications are often long, and the authors may well have begun their project before the Margoni & Surian meta analysis was published. Likewise, Woo & Hamlin may have only disclosed the positive 12-month-old results, so the authors may not have known about the null results with 8- and 10-month-old infants. If that's the case, it's fine for them to say that they selected their procedure based on the evidence/recommendation they had at the time of study design. However, I would recommend including stronger emphasis on (or at least some mention of) the within-lab discrepancy between results of live-action and animated stimuli obtained by the original authors. They could also enumerate other potential reasons, beyond screen-to-object mapping, such a discrepancy would exist (e.g. animated stimuli fail to elicit same goal attribution to protagonist, especially in 2D environment where there may be no prior that moving upward is difficult? necessarily non-blind puppeteers have unintended experimenter effects on preferences?).

Author's Response to Decision Letter for (RSOS-191795.R2)

See Appendix D.

Decision letter (RSOS-191795.R3)

04-Mar-2020

Dear Ms Schlingloff:

It is a pleasure to accept your manuscript entitled "Do 15-month-old infants prefer helpers? A replication of Hamlin et al. (2007)" in its current form for publication in Royal Society Open Science.

on behalf of Professor Chris Chambers (Subject Editor)
openscience@royalsociety.org

Appendix A

Laura Schlingloff
Cognitive Development Center, Department of Cognitive Science
Central European University
Október 6 utca 7
1051 Budapest
Hungary

11.October 2019

Dear Editor,

We are hereby resubmitting a revised version of our Stage 1 Replication manuscript, entitled “Do 15-month-old infants prefer helpers? A replication of Hamlin et al. (2007)”, by Laura Schlingloff, Gergely Csibra, and Denis Tatone (manuscript ID: RSOS-191747). The target article for this replication is Hamlin, J. K., Wynn, K. & Bloom, P. Social evaluation by preverbal infants. *Nature* **450**, 557-559 (2007).

As requested by the Associate Editor, we included in the “Coding and Analyses” section a description of the exact analysis that will be conducted with the data.

Please address correspondence regarding this manuscript to:
schlingloff_laura@phd.ceu.edu.

Thank you for your consideration. We are looking forward to your response.

Sincerely,

Laura Schlingloff, Gergely Csibra, Denis Tatone

Appendix B

RSOS-191795: “Do 15-month-old infants prefer helpers? A replication of Hamlin et al. (2007)”

Response to Referees:

R1/R2:

- Justify the sample size if it was based on the sample size in Woo & Hamlin, then say so)

- Can the authors provide a power analysis outlining their expectations about effect size and justifying intended sample size? Going off the previous results w/ the same stimulus set (23/32), what are odds of observing effect in new sample of 32? For high certainty of replicating a true effect (which seems like the goal here), 2.5x original sample has sometimes been recommended (Simonsohn, 2015)... the preregistered one-tailed test does help, but I would suggest at least extending the sample to 1.5x the original size of Woo & Hamlin (N = 48), so that the outcome of the study is more convincing to readers who come in with preconceived notions about the status of this effect from either direction.

Our study was intended as a replication and extension of Hamlin et al. (2007) modeled after a previous replication attempt by Woo and Hamlin (unpublished), which found a preference for helpers with infants from 12 months of age (Hamlin, personal communication). For this reason, we decided to double the sample size of the original study. It is worth noting that many studies in the field of infant social evaluation have sample sizes of 20 or fewer subjects and studies with 32 or more as the exception (see Margoni & Surian, 2018). We believe that this sample size should provide us with sufficient power to reject the null hypothesis of no effect (which, unlike a null of detectable effects, does not necessarily require larger samples to be rejected: Simonsohn, 2015).

Nevertheless, we will qualify the results of our study by performing and reporting a Bayesian analysis (reported in the paper in an “Exploratory Results” section, as this analysis was not performed by the authors of the original study), which allows us to evaluate the strength of the evidence for supporting either H0 (no effect) or H1.

R1:

- The way it is written now is that the interpretations of the original authors are the only interpretation of a reaching preference for helpers. This work is undoubtedly useful for the field. However, if the authors find the same effect, it does not necessarily mean that the original interpretation of the data is the only plausible one.

Independent of the outcome, our results cannot speak to potential different interpretations of the original findings (i.e., whether infants prefer “helper” characters because they engage in sociomoral evaluation, or whether their choice behavior is driven by other features

of the stimuli), as our study is not designed to disambiguate between them. In the discussion section of our paper, we will frame the result accordingly and mention that it should not be taken as adjudicating on potential cognitive mechanisms.

R2:

- The description of parent instructions in the first paragraph of Procedure section is a little hard to follow; I initially interpreted it as parents were instructed to have their chair turned away from the screen but their head/body facing the screen throughout the study. Instead, I think the authors meant that after training on the choice phase, parents were asked to turn their chair back toward the screen for the familiarization phase. Wording along these lines might prevent my (possibly silly) misinterpretation.

This is indeed what we meant. We thank the reviewer for the suggestion. We have updated the section accordingly.

R2:

- What if an infant reaches out and touches one character while looking elsewhere, e.g. at the other character or the experimenter? Is that infant excluded? Are any attempts made to discourage this and elicit a valid reach, e.g. by pulling board back if a baby is reaching without looking, and encouraging the babies to look at the characters before extending the board again?

If an infant touches a character without looking at it prior to and during the reach, this is not coded as a visually-guided reach. In such a case, Experimenter 2 allows the infant additional time for choosing while keeping the board within his reach and providing verbal encouragement; if the infant fails to produce a valid reach within the 2 minute window for the manual choice phase, this infant is excluded from the sample as per Hamlin's criteria. Hamlin recommended that we do not pull the board back after it has been flipped over while the infant is reaching toward it, as this might signal to the infant that the choice he was about to make is "wrong". We have added some clarifications on this at the penultimate paragraphs of the "Procedure" section, and in the first paragraph of the "Coding and Analyses" section.

R2:

- How will coder reliability be assessed? Does Exp 2 also judge and record the infant's response after administering the choice procedure?

The experimenter who administers the choice procedure only judges the response insofar as to make a decision on-line whether to terminate the choice phase (i.e., whether the infant has made a clear choice). Responses are recorded (1) by Experimenter 1, and (2) by an independent second coder who is not present while the experiment is run and who is blind to the experimental condition (identity of the Helper/Hinderer). In cases of disagreement (e.g., when the infant's choice is judged by the second coder to be ambiguous), the participant

is removed from the final sample and replaced. We have updated the second paragraph in the “Coding and Analyses” section to clarify this point.

R2:

- The familiarization looking inclusion criterion seems strict, especially for this age range. What if an infant extensively habituates after substantial looking over the first 4 trials? Is this criterion following Hamlin & Woo? If not, I would encourage relaxing criterion to 50% of trial for at least 2 familiarization trials of each type, rather than all three. This better matches what we would ask of an infant in a habituation procedure, and it has already been noted that this age range is often too impatient for those.

Most studies of this type use a habituation procedure (albeit with younger infants) where infants typically habituate in 8-9 trials. We wanted to ensure that our participants would still be exposed to a sufficient number of the crucial helping and hindering events. Considering the fact that we only present them with 3 repetitions of each event, we decided to apply this strong inclusion criterion, as we assume that manual choice data from inattentive babies would not be informative. We will make transparent the choice data from participants who are excluded due to failing to meet the criterion, and will report whether including these subjects changes the results.

Appendix C

Do 15-month-old infants prefer helpers? A replication of Hamlin et al. (2007)

Laura Schlingloff^{*1}

schlingloff_laura@phd.ceu.edu

Gergely Csibra^{1,2}

CsibraG@ceu.edu

Denis Tatone¹

denis.tatone@gmail.com

¹ Cognitive Development Center, Department of Cognitive Science, Central European University, Budapest, Hungary

² Department of Psychological Sciences, Birkbeck, University of London, UK

Abstract

Hamlin et al. found in 2007 that preverbal infants displayed a preference for helpers over hinderers. The robustness of this finding and the conditions under which infant sociomoral evaluation can be elicited has since been debated. Here, we conducted a replication of the original study, in which we tested 14- to 16-month-olds using a familiarization procedure with 3D-animated video stimuli. Unlike previous replication attempts, ours uniquely benefitted from detailed procedural advice by Hamlin. In contrast to the original results, only 16 out of 32 infants (50%) in our study reached for the helper; thus, we were not able to replicate the findings. A possible reason for this failure is that infants' preference for prosocial agents may not be reliably elicited with the procedure and stimuli adopted. Alternatively, the effect size of infants' preference may be smaller than originally estimated. The study addresses ongoing methodological debates on the replicability of influential findings in infant cognition.

Keywords: social evaluation, infant cognition, manual choice paradigm, moral judgment, partner choice

Introduction

A growing literature suggests that, from a very young age, infants spontaneously engage in third-party social evaluation, drawing inferences about the sociomoral dispositions of unrelated agents on the basis of their interactions with others. This proliferating research project was launched by the seminal 2007 study of Hamlin et al.¹, which showed that 6- and 10-month-olds presented with two characters interacting in a helpful or harmful manner towards a common patient subsequently preferred the former when prompted to choose among the two.

This study, and others that followed in its wake, bolstered the empirical case for an “innate moral core”²: an early-developing set of abilities that allows infants to infer sociomorally relevant dispositions from the behavior of third parties, which in turn enables infants to recognize and selectively interact with potentially cooperative partners. Support for this account has been found in a variety of “morality plays”. Besides the original 2007 study, which featured a character attempting to climb up the hill and being pushed up or down respectively by two other characters (‘hill’

paradigm), infants' preference for prosocial agents has been explored in a number of instrumental helping scenarios: one requiring a box to be opened to retrieve a desired object ('box' paradigm³); another requiring a ball to be returned to its owner, who accidentally dropped it while playing ('ball' paradigm³); others requiring a shelf to be knocked or a door to be opened to make an out-of-reach object accessible^{4,5}.

Adding to the situational breadth of early social evaluation that these studies attested, others showed this to be a nuanced and sophisticated phenomenon. Already in their first year of life, infants appear sensitive to epistemic states and overt intentions: they prefer intentional over accidental Helpers, but accidental over intentional Hinderers⁴, and unsuccessful Helpers over unsuccessful Hinderers⁵. Additionally, infants show a preference for Helpers only when these know the particular goal the Helpee is trying to accomplish^{6,7}. Moreover, infants do not choose characters on the basis of the mere valence of the actions they performed, but interpret them in context, preferring a character who 'punishes' – i.e., acts antisocially towards – a previous Hinderer over a character who helps her^{8,9}.

Beyond instrumental helping, a preference for prosocial characters has been found in a number of other sociomoral domains. In the domain of physical aggression, for instance, infants preferred victims over perpetrators^{10,11}, and third-party characters intervening in a conflict to shield victims from ongoing aggression over passive bystanders¹². Similarly, in the domain of resource allocation, infants have been repeatedly shown to prefer fair distributors over unfair ones^{13,14,15}. Modified versions of the manual-choice paradigm have also been recently used to investigate whether similar evaluative tendencies exist in non-human animals, such as bonobos¹⁶, capuchin monkeys¹⁷, and dogs¹⁸.

Despite the recent growth of the literature on early sociomoral evaluation, attempts to replicate the findings by Hamlin and colleagues have yielded mixed results. For example, using the original "hill scenario", Cowell and Decety¹⁹ found no significant preference for Helpers in 12-to-24-month-olds (see also Colaizzi²⁰). Similarly, Scarf et al.²¹ suggested that low-level perceptual features, rather than inferred sociomoral dispositions, could adequately explain infants' preference for prosocial characters (though see Hamlin⁶ for a rebuttal of this claim). Using the "box scenario", Salvadori et al.²² found no preference for Helpers across two experimental attempts. A similar lack of preference was documented by Nighbor et al.²³ with 5- to 16-month-olds. Conversely, using the "ball scenario", Scola et al.²⁴ reported a significant preference for prosocial characters in 12- to 24- and 24- to 36-month-olds, whereas Shimizu et

al.²⁵ documented a similar, albeit weaker, preference in 15- to 18-month-olds, but not in younger age groups. It is worth noting, however, that previous replication attempts have followed the methods of the original studies to varying degrees of fidelity. Differences in stimuli materials and procedural details might have conceivably affected infants' responses.

In a recent meta-analysis, Margoni and Surian²⁶ reviewed 26 published and unpublished studies using manual-choice measures to investigate early sociomoral evaluation. While their analysis revealed an overall significant tendency to prefer prosocial characters across studies, the authors cautioned about the possibility of publication bias and the underreporting of negative findings (file drawer problem). Importantly, Margoni and Surian also attested the presence of a lab effect: research conducted by Hamlin and collaborators tends to generate larger effect sizes compared to studies done by independent laboratories. On these grounds, the authors called for more and sufficiently powered replications.

Here we conducted a conceptual replication of the original study by Hamlin et al.¹. Our study differs from the original in three potentially important ways. Firstly, we tested 15-month-old infants, an age group slightly older than the infants tested in similar studies. While Margoni and Surian's meta-analysis²⁶ found no significant effect of age on infants' preference for prosocial characters, the participants' mean age in the studies reviewed was approximately 13 months (390 days). Secondly, we did not present the stimuli in the form of a live puppet show, but as video animations on a screen, which were generously provided to us by Woo and Hamlin. Although Margoni and Surian²⁶ found no effect of presentation mode (live vs. video), a majority of the studies in their sample were based on live puppet shows. Thirdly, instead of using a habituation procedure, we employed a familiarization procedure, presenting the stimuli for a fixed amount of time across infants. This was aimed at mitigating the problem of fussiness and high drop-out rates, common with older infants when using habituation designs.

Crucially, these modifications were implemented under recommendation of Woo and Hamlin, who used the same video stimuli and familiarization procedure for their own in-lab replication (in preparation) of the original Hamlin et al. study¹. In said replication, Woo and Hamlin found a significant preference for the helper character in a sample of 32 infants (23 of 32; reported in Margoni and Surian's meta-analysis²⁶).

Methods

This article received results-blind in-principle acceptance (IPA) at Royal Society Open Science. Following IPA, the accepted Stage 1 version of the manuscript, not including results and discussion, was preregistered on the OSF (<https://osf.io/krms8>). The preregistration was produced after data collection and analysis.

Piloting phase

Before testing our experimental sample, we conducted a pilot with 24 infants aged 14 to 16 months. During the piloting phase, we sent video recordings of the participants to Hamlin (written permission for data sharing was obtained from the parents), who kindly provided helpful feedback on the procedure, and we subsequently implemented her suggestions. Testing of the experimental sample began only after Hamlin had confirmed that our procedure was sufficiently close to the original.

Participants

Thirty-two 14- to 16-month-old infants participated in the study (mean age: 15.18 m, range: 431-492 d, 20 males). The sample size was determined prior to data collection and was twice the sample of 10-month-olds and more than twice the sample of 6-month-olds tested in Hamlin et al. (2007). An additional 19 infants were tested but not included in the final sample due to failing to produce a choice at test ($n = 7$), inattentiveness during familiarization ($n = 5$), fussiness ($n = 4$), experimenter error ($n = 2$), and technical failure ($n = 1$). Participants were full-term infants with no reported health or developmental issues. Infants were recruited from the database of the Cognitive Development Center, which includes contact information of parents volunteering to participate in research with their children. Data collection took place between January and May 2018.

Ethical approval

Caregivers were informed about the nature and possible consequences of the study, and gave informed consent for their child to participate. We obtained ethical approval

for this research from the United Ethical Review Committee for Research in Psychology (EPKEB) in Hungary, and it was conducted according to the ethical rules and standards regarding psychological experimentation in Hungary.

Materials and apparatus

During the familiarization phase, infants were seated in their caregiver's lap in a dimly lit room, approximately 60 cm away from a TV screen of 100 cm diagonal size. The stimuli were generated by Woo and Hamlin using Blender animation software (<https://www.blender.org/download>), and were presented on a screen using PsyScope X²⁷ controlled by a Mac Mini computer.

The objects for the manual choice procedure were printed out versions of the blue square and yellow triangle characters from the stimuli videos (square: 13x13 cm, triangle: 15.5x13.5 cm). The printout graphics were converted from RGB to CMYK color space and adjusted, so that the color of the printed characters matched those on screen as closely as possible. Printouts were glued onto figures made of stacked cardboard, to mimic the three-dimensional appearance of the characters in the video. The figures were then wrapped with a transparent plastic cover, to protect them from wear. The figures were attached with removable adhesive putty onto a white board (50x36 cm) at a distance of 19 cm from each other, 3 cm from the sides of the board, and 3 cm from the bottom of the board.

During the familiarization, Experimenter 1, who ran the study and coded the infants' looking behavior online, was seated in the same room as the child, hiding behind a black curtain. Experimenter 2, who performed the manual choice task, also hid behind the curtain during the familiarization phase. To ensure that Experimenter 2 was blind to condition, she had no visual access to the screen displaying the stimuli.

Procedure

Before the familiarization phase, Experimenter 2 briefed the caregiver on how to position herself for the manual choice task. The caregiver was instructed to turn her chair away from the screen, place her feet behind a tape marking on the floor, and have the child sit on her knees while supporting the child by the ribcage. After this training on the choice phase, the caregiver was asked to turn back towards the

screen for the familiarization phase and to keep her eyes closed for the whole duration of the study.

Familiarization phase. Infants watched a total of six familiarization trials featuring three helping and three hindering events, alternated. Each trial was preceded by a brief attention-getter (a flashing checkerboard accompanied by the sound of a xylophone slide) which played until the child gazed back at the screen. The two familiarization events were matched in timing and overall duration (17 s).

Both events took place on a light-blue sky background and a dark green hill, extending from the bottom left to the top right corner of the screen. The hill plateaued halfway and at the top.

Each event started with a character (a small red circle with eyes pointing to the top of the hill; hereinafter, Protagonist) located at the bottom of the hill. After a bell sound, the Protagonist moved to the intermediate plateau and bounced up and down twice with her eyes directed towards the viewer (2 s). She then attempted to climb the top plateau twice, each time reaching up to two-thirds of the steep incline and sliding back down to the intermediate plateau (8 s). At this point the Helper or Hinderer appeared, again to the sound of a bell (Helper: from the bottom left of the screen; Hinderer: from the top right of the screen). As the Protagonist attempted to climb the steep incline to the top plateau for a third time, the Helper/Hinderer (whose eyes were directed to the top or bottom of the hill, respectively) moved towards the Protagonist and, with two repeated shoves (accompanied by a knocking sound), pushed the Protagonist up to the top plateau or down to the bottom one (4 s). Upon reaching either the bottom or the top of the hill, the Protagonist came to a standstill, while the other character exited the scene from the location where she initially appeared (3 s).

Each trial ended with a still frame, kept on screen until the infants had looked away for a minimum of two consecutive seconds or until 30 seconds had elapsed.

Test phase. Immediately after the end of the familiarization phase, the screen turned black and a soft guitar tune started playing (also provided by Woo and Hamlin). Experimenter 2, following a cue from Experimenter 1, entered the testing room from behind the curtain, turned on the light, and instructed the caregiver to assume the previously practiced position for the manual choice task and to close her eyes again afterwards. Experimenter 2 knelt down in front of the child and addressed her while making eye contact: “*Szia [name of child]! Kivel szeretnél játszani?*”, which

translates to *“Hi [name of child]! Who would you like to play with?”*. Then, she lowered her gaze to the chin of the child and flipped over the board with the two characters. The board was moved towards the infant and turned slightly downward at approximately a 30-degree angle, so that the figures were within the infant’s reach but required participants to stretch out their arms in order to touch them. After the board had been flipped over, the experimenter made sure not to pull the board away while the infant was reaching out for a character, as this might convey to the infant that her intended choice was “wrong” (Hamlin, personal communication).

If the infant did not produce any visually-guided reaching after about 30 s, Experimenter 2 verbally encouraged the infant by saying, for instance, “Figyelj!” (“Pay attention!”), “Nézd meg őket!” (“Look at them!”), or “Bátran!” (“Be brave!”), and repeating the original question. If no choice was produced after two minutes, the experiment was terminated.

The following factors were counterbalanced in the study: the identity of Helper and Hinderer during familiarization (blue square vs. yellow triangle), the order of event presentation (helping first vs. second), and the position of the characters on the board (helper on the right vs. left side). The condition that each infant was assigned to was randomly selected before testing.

Coding and Analyses

The dependent variable was the infants’ choice of the Helper or Hinderer character, assessed by their reaching to one of the figures on the board. In order to be counted as a choice, the reaching had to be visually guided: i.e., infants had to look at a character before and while touching it. If infants reached for a figure while looking elsewhere, they were given the opportunity to produce another reach within the 2 minute time window. If infants touched both figures, but looked only at one prior to establishing contact, this was coded as a choice for the figure they looked at.

Experimenter 2 judged on-line whether the infant had reached unambiguously for one of the figures and thus whether to terminate a trial. Choices were coded off-line from the videos by Experimenter 1, and recoded by an independent second coder blind to the experimental condition, reaching 93.75% of agreement. Two infants judged by the second coder to have made no clear choice were removed from the final sample and replaced.

In order to be included in the final data analysis, infants had to watch at least 50% of the duration of each helping/hindering event (from the onset of physical contact between the protagonist and the Helper/Hinderer to the end of the pushing action) in all six trials. This stringent criterion of attentiveness was meant to ensure that each infant attended to the crucial social interactions differentiating Helper and Hinderer for a sufficient number of times. Including the manual choice data from the infants who did not meet this criterion did not affect the results.

In order to assess whether infants showed preference for the Helper character, we performed a one-tailed binomial test on the number of infants who chose the Helper and the total number of infants included in the analysis against the probability of 0.5 as chance level (as was done by Hamlin et al., 2007). Statistical analyses were performed with R, the lme4 package²⁸, and the BayesFactor package²⁹. Data are available at <https://osf.io/kh5r4/>.

Results

Hypothesis-driven analyses

Sixteen out of 32 children directed their first visually guided reach to the Helper (one-tailed binomial test $p = .57$; 95% CI: .344-1.0). Thus, infants did not display a preference for either the Helper or the Hinderer character. When including in the analyses the 5 infants who were excluded due to inattentiveness during the familiarization phase, 20 out of 37 reached for the Helper (one-tailed binomial test $p = .371$; 95% CI: .394-1.0).

Further results

In a Bayesian analysis with a null model of $p = .5$ and an alternative model with a uniform prior (implemented in the BayesFactor package by an “ultrawide” scale parameter of 1), the data from our study yielded a Bayes factor of 4.618 in favor of H_0 , indicating moderate support for the null hypothesis of no effect²⁹.

Infants' choice was not significantly influenced by their gender (9 of 20 male infants chose the helper, while 7 of 12 females did), characters' features (20 of 32 infants reached for the yellow triangle), characters' location on the board (17 infants reached

for the figure on the right), and order of familiarization events (12 infants reached for the agent they last saw).

During the manual choice, a subset of infants did not unambiguously direct their gaze at both characters before producing a choice. This, however, did not affect the results: 12 of 24 of those infants who looked at both characters chose the Helper, whereas 4 of 8 of those who only looked at one character reached for the Helper.

We also analyzed whether the amount of looking to the two types of familiarization events may have influenced the infants' behavior at test. In line with previous studies, we found no difference in mean looking times to the still frames following the two events (helping: 11.41 s; hindering: 11.39 s). We fit a mixed effects linear regression model predicting log looking time from familiarization event type with a subject-random intercept. Model comparison revealed no significant looking time difference between the event types (Chi-squared = 0.0294, $p = .864$). Moreover, infants did not tend to choose the agent they attended to longer on average during familiarization (16 of 32 reached for the character they had looked at longer).

Discussion

In the present study we attempted to replicate Hamlin et al.'s (2007) finding that infants preferentially reach for helpful over hindering characters¹. In that study, 92.9% of infants exhibited such preferences (14 of 16 10-month-olds and 12 of 12 6-month-olds). In contrast, only 50% of infants did so in our study (16 of 32). Therefore, we could not reproduce the original findings. There are several potential explanations for such a failure. Our study differed from the original in three potentially relevant ways: firstly, we tested infants from an older age group (15-month-olds, rather than 6- and 10-month-olds); secondly, we used 3D-animated videos rather than a live puppet show to expose infants to the helping and hindering events; and thirdly, we used a familiarization rather than a habituation design.

Any of these deviations from the original study may have potentially contributed to our results. For instance, it is conceivable that six familiarization trials were insufficient for infants to learn about the agents' respective dispositions. Supporting this possibility, average looking times decreased from 12.81 s in the first three trials to 9.99 s in the last three trials, which constitutes a decrease of 22% in looking, hence insufficient to reach the habituation criterion adopted by Hamlin (decrease of

50% from the first three to the last three trials). It should be however noted that prior studies⁷ and the in-lab replication onto which our study was modelled successfully elicited a preference for helpers by means of familiarization.

Alternatively, infants may have had troubles mapping the cardboard replicas of Helper and Hinderer to the 3D-animated characters they were familiarized to. However, several studies reported preferential reaching for replicas of prosocial characters presented on screen^{24, 30, 31}. More importantly, the use of computer animations as well as the length and type of exposure that infants received prior to the manual choice was determined on the basis of Hamlin & Woo's recommendation and results of their own replication. Thus, we had no reason to suspect in advance that these factors would affect infants' behaviors.

It is also possible that other unforeseen methodological differences, some of which may be hard or impossible to control for, contributed to our failed replication. Such differences may concern, for instance, the physical set-up of the testing environment, the cultural background of the population tested, or, more likely, the behavior of the experimenters involved in the study. On this note, it is however worth noting that, unlike other replication attempts, ours benefitted from the close and careful scrutiny of the experimenters' behavior by Hamlin herself. Her feedbacks during the piloting phase allowed us to fine-tune the procedure of character presentation in ways that other studies could not.

Finally, current evidence suggests that the underlying effect size of infants' preference for helpful characters may be smaller than originally assumed. The meta-analysis by Margoni and Surian²⁶ estimated that on average 64% of infants in the studies reviewed reached for the prosocial character. Importantly, however, the strength of infants' preference was affected by the sociomoral domain tested: 77% of infants preferred the prosocial character after observing giving vs. taking events, 69% after observing fair vs. unfair distributions, and only 63% after observing helping vs. hindering events. Although instrumental helping represented the domain with the lowest percentage of infants' choice of the prosocial agent, this was nevertheless considerably higher than the percentage (50%) obtained in our study.

In a recent paper, Margoni and Shepperd³² argued that individual replication studies ought not to be considered as confirming or disconfirming an effect, but rather should be pooled together to produce a better estimate of the true underlying effect size of the phenomenon at hand. If original studies are underpowered, as is often the case

in infant research, replications with a relatively wide range of results may technically be taken as confirming the original finding if they fall within a “prediction interval” of potential outcomes. This said, our proportion of 50% helper choices falls outside the value range (.59-1.0) defined by the 95% prediction interval proposed by Margoni and Shepperd for a replication of Hamlin et al.’s (2007) study with $n = 32$, and thus cannot be considered confirmatory.

The present replication sheds further light on the robustness of the phenomenon of early sociomoral evaluation and the conditions under which it can be reliably elicited. It also contributes to broader methodological debates on the replicability of findings in developmental science, and reaffirms the need, already voiced by Margoni & Surian, for multi-lab replication initiatives³³ that could adequately assess the influence of potentially mediating factors.

References

- 1 Hamlin, J. K., Wynn, K. & Bloom, P. Social evaluation by preverbal infants. *Nature* **450**, 557-559 (2007). doi: 10.1038/nature06288
- 2 Hamlin, J. K. Moral judgment and action in preverbal infants and toddlers: Evidence for an innate moral core. *Curr. Dir. Psychol. Sci.* **22**, 186-193 (2013). doi: 10.1177/0963721412470687
- 3 Hamlin, J. K. & Wynn, K. Young infants prefer prosocial to antisocial others. *Cogn. Dev.* **26**, 30-39 (2011). doi: 10.1016/j.cogdev.2010.09.001
- 4 Woo, B. M., Steckler, C. M., Le, D. T. & Hamlin, J. K. Social evaluation of intentional, truly accidental, and negligently accidental helpers and harmers by 10-month-old infants. *Cognition* **168**, 154-163 (2017). doi: 10.1016/j.cognition.2017.06.029
- 5 Hamlin, J. K. Failed attempts to help and harm: Intention versus outcome in preverbal infants' social evaluations. *Cognition* **128**, 451 – 474 (2013). doi: 10.1016/j.cognition.2013.04.004
- 6 Hamlin, J. K. The case for social evaluation in preverbal infants: gazing toward one's goal drives infants' preferences for Helpers over Hinderers in the hill paradigm. *Front. Psychol.* **5**, 1563 (2015). doi: 10.3389/fpsyg.2014.01563
- 7 Hamlin, J. K., Ullman, T., Tenenbaum, J., Goodman, N. & Baker, C. The mentalistic basis of core social cognition: experiments in preverbal infants and a computational model. *Dev. Sci.* **16**, 209 – 226 (2013). doi: 10.1111/desc.12017
- 8 Hamlin, J. K., Wynn, K., Bloom, P. & Mahajan, N. How infants and toddlers react to antisocial others. *Proc. Natl. Acad. Sci. U.S.A.* **108**, 19931-19936 (2011). doi: 10.1073/pnas.1110306108
- 9 Hamlin, J. K. Context-dependent social evaluation in 4.5-month-old human infants: The role of domain-general versus domain-specific processes in the development of social evaluation. *Front. Psychol.* **5**, 614 (2014). doi: 10.3389/fpsyg.2014.00614

- 10** Kanakogi, Y., Okumura, Y., Inoue, Y., Kitazaki, M. & Itakura, S. Rudimentary sympathy in preverbal infants: preference for others in distress. *PLOS ONE* **8**, e65292 (2013). doi: 10.1371/journal.pone.0065292
- 11** Uzefovsky, F., Paz, Y. & Davidov, M. Young infants are pro-victim, but it depends on the context. *British Journal of Psychology* (in press). doi: 10.1111/bjop.12402
- 12** Kanakogi, Y., Inoue, Y., Matsuda, G., Butler, D., Hiraki, K. & Myowa-Yamakoshi, M. Preverbal infants affirm third-party interventions that protect victims from aggressors. *Nat. Hum. Behav.* **1**, 0037 (2017). doi: 10.1038/s41562-016-0037
- 13** Burns, M. P. & Sommerville, J.A. "I pick you": the impact of fairness and race on infants' selection of social partners. *Front. Psychol.* **5**, 93 (2015). doi: 10.3389/fpsyg.2014.00093
- 14** Geraci, A. & Surian, L. The developmental roots of fairness: infants' reactions to equal and unequal distributions of resources. *Dev. Sci.* **14**, 1012-1020 (2011). doi: 10.1111/j.1467-7687.2011.01048.x
- 15** Lucca, K., Pospisil, J. & Sommerville, J. A. (2018). Fairness informs social decision-making in infancy. *PLOS ONE* **13**, e0192848 (2018). doi: 10.1371/journal.pone.0192848
- 16** Krupenye, C. & Hare, B. Bonobos prefer individuals that hinder over those that help. *Curr. Biol.* **28**, 280-286 (2018). doi: 10.1016/j.cub.2017.11.061
- 17** Anderson, J. R., Kuroshima, H., Takimoto, A. & Fujita, K. Third-party social evaluation of humans by monkeys. *Nat. Commun.* **4**, 1561 (2013). doi: 10.1038/ncomms2495
- 18** McAuliffe, K., Bogese, M., Chang, L. W., Andrews, C. E., Mayer, T., Faranda, A., Hamlin, J. K. & Santos, L. R. Do Dogs Prefer Helpers in an Infant-Based Social Evaluation Task? *Front. Psychol.* **10**, 591 (2019). doi: 10.3389/fpsyg.2019.00591

- 19** Cowell, J. M. & Decety, J. Precursors to morality in development as a complex interplay between neural, socioenvironmental, and behavioral facets. *PNAS* **112**, 12657-12662 (2015). doi: 10.1073/pnas.1508832112
- 20** Colaizzi, J. M. Empathy and Prosocial Behaviors in Infancy. PhD thesis at Oklahoma State University; https://shareok.org/bitstream/handle/11244/48793/Colaizzi_okstate_0664D_14781.pdf?sequence=1 (2016).
- 21** Scarf, D., Imuta, K., Colombo, M. & Haye, H. Social Evaluation or Simple Association? Simple Associations May Explain Moral Reasoning in Infants. *PLOS ONE* **7**, e42698 (2012). doi: 10.1371/journal.pone.0042698
- 22** Salvadori, E. *et al.* Probing the Strength of Infants' Preference for Helpers over Hinderers: Two Replication Attempts of Hamlin and Wynn (2011). *PLOS ONE* **10**, e0140570 (2015). doi: 10.1371/journal.pone.0140570
- 23** Nighbor, T., Kohn, C., Normand, M. & Schlinger, H. Stability of infants' preference for prosocial others: Implications for research based on single-choice paradigms. *PLOS ONE* **12**, e0178818 (2017). doi: 10.1371/journal.pone.0178818
- 24** Scola, C., Holvoet, C., Arciszewski, T. & Picard, D. Further Evidence for Infants' Preference for Prosocial Over Antisocial Behaviors. *Infancy* **20**, 684-692 (2015). doi: 10.1111/inf.12095
- 25** Shimizu, Y., Senzaki, S. & Uleman, J. S. The Influence of Maternal Socialization on Infants' Social Evaluation in Two Cultures. *Infancy* **23**, 748-766 (2018). doi: 10.1111/inf.12240
- 26** Margoni, F. & Surian, L. Infants' evaluation of prosocial and antisocial agents: A meta-analysis. *Dev. Psychol.* **54**, 1445-1455 (2018). doi: 10.1037/dev0000538
- 27** Cohen, J. D., MacWhinney, B., Flatt, M. & Provost, J. PsyScope: A new graphic interactive environment for designing psychology experiments. *Behav. Res. Meth. Instrum. Comput.* **25**, 257-271 [Computer software]; retrieved from <http://psy.ck.sissa.it> (1993).

- 28** Bates, D., Maechler, M., Bolker, B. & Walker, S. Fitting Linear Mixed-Effects Models Using lme4. *Journal of Statistical Software* **67**, 1-48 [Computer software]; retrieved from <https://cran.r-project.org/web/packages/lme4/index.html> (2015).
- 29** Morey, R. D. & Rouder, J. N. BayesFactor: Computation of Bayes Factors for Common Designs. R package version 0.9.12-4.2 [Computer software]; retrieved from <https://CRAN.R-project.org/package=BayesFactor> (2018).
- 30** Thomas, A. J. & Sarnecka, B. W. Infants Choose Those Who Defer in Conflicts. *Curr. Biol.* **29**, 2183-2189 (2019).
- 31** Powell, L. J. & Spelke, E. S. Third-Party Preferences for Imitators in Preverbal Infants. *Open Mind* **2**, 61-71 (2018).
- 32** Margoni, F. & Shepperd, M. Changing the logic of replication: A case study from infant studies. Preprint retrieved from <https://psyarxiv.com/xw6qt/>.
- 33** Frank, M. C. *et al.* A collaborative approach to infant research: Promoting reproducibility, best practices, and theory-building. *Infancy* **22**, 421-435 (2017).

Data accessibility

The dataset generated and analyzed during the current study is available at the OSF repository (<https://osf.io/kh5r4/>), as are the stimuli (shared with permission from Brandon Woo and Kiley Hamlin).

Acknowledgments

We thank K. Hamlin and B. Woo for help with implementing the experimental procedure and providing the stimuli, M. Nagy for assistance with data collection, and P. Rácz for statistical advice.

Funding

This research has received partial funding from the European Research Council (ERC) under the European Union's Horizon 2020 research and innovation programme under grant agreement No #742231 ("PARTNERS").

Author Contributions

L. S. performed research and analyzed the data; L.S., G.C. and D.T. wrote the paper.

Competing Interests

The authors declare no competing interests.

Appendix D

RSOS-191795.R2: “Do 15-month-old infants prefer helpers? A replication of Hamlin et al. (2007)”

Response to Reviewers:

R1:

- *Thomas & Sarnecka presented live puppet shows to infants, not animations.*

We thank the Reviewer for having made us aware of our mistake. We edited the references in the third paragraph of the “Discussion” section accordingly.

R2:

- *The authors include a few exploratory analyses of factors that may have affected infants’ helper preferences. One they did not include but that may be of interest given the overall weak level of habituation across the six familiarization is to ask if stronger habituation (i.e. greater proportional decrease in looking from the first three trials to the last three) predicted a higher likelihood of reaching for the helper.*

We added the requested analysis, and moved the information about infants’ habituation pattern, to the “Further results” section.

R2:

- *[...] The timelines of research projects and publications are often long, and the authors may well have begun their project before the Margoni & Surian meta analysis was published. Likewise, Woo & Hamlin may have only disclosed the positive 12-month-old results, so the authors may not have known about the null results with 8- and 10-month-old infants. If that’s the case, it’s fine for them to say that they selected their procedure based on the evidence/recommendation they had at the time of study design. However, I would recommend including stronger emphasis on (or at least some mention of) the within-lab discrepancy between results of live-action and animated stimuli obtained by the original authors.*

We greatly thank the Reviewer for this thoughtful critique. We did plan our study before Margoni and Surian’s meta-analysis was published, on the basis of recommendations by Hamlin that the procedure was well-suited for infants beyond 12 months of age (as their in-lab replication showed). Data collection for the younger age groups in Woo and Hamlin’s study was ongoing at the time we started our replication. However, both the lower percentage of infants choosing the helper in their in-lab replication relatively to Hamlin et al. (2007) and the failure to elicit such preference in two samples of younger infants give plausibility

to the claim that differences in stimuli and design between the original study and the present replication may contribute to explain the divergent findings. To address the Reviewer's concern, we introduced a new paragraph in the "Discussion" section to make these differences explicit.